# Partially Frozen Random Networks Contain Compact Strong Lottery Tickets

**Hikari Otsuka**[1][*][†]                                  *otsuka.hikari@artic.iir.isct.ac.jp*

**Daiki Chijiwa**[2][*]                                       *daiki.chijiwa@ntt.com*

**Ángel López García-Arias**[2][*]                              *lopez@ieee.org*

**Yasuyuki Okoshi**[1]                                  *okoshi.yasuyuki@artic.iir.isct.ac.jp*

**Kazushi Kawamura**[1]                                  *kawamura@artic.iir.isct.ac.jp*

**Thiem Van Chu**[1]                                       *thiem@artic.iir.isct.ac.jp*

**Daichi Fujiki**[1]                                         *dfujiki@artic.iir.isct.ac.jp*

**Susumu Takeuchi**[2]                                     *susumu.takeuchi@ntt.com*

**Masato Motomura**[1]                                    *motomura@artic.iir.isct.ac.jp*

[1]*Department of Information and Communications Engineering, Institute of Science Tokyo, Japan*
[2]*NTT Corporation, Japan*

**Reviewed on OpenReview:** *https://openreview.net/forum?id=xpnPYfufhz*

## Abstract

Randomly initialized dense networks contain subnetworks that achieve high accuracy without weight learning—strong lottery tickets (SLTs). Recently, Gadhikar et al. (2023) demonstrated that SLTs could also be found within a randomly pruned source network. This phenomenon can be exploited to further compress the small memory size required by SLTs. However, their method is limited to SLTs that are even sparser than the source, leading to worse accuracy due to unintentionally high sparsity. This paper proposes a method for reducing the SLT memory size without restricting the sparsity of the SLTs that can be found. A random subset of the initial weights is frozen by either permanently pruning them or locking them as a fixed part of the SLT, resulting in a smaller model size. Experimental results show that `Edge-Popup` (Ramanujan et al., 2020; Sreenivasan et al., 2022) finds SLTs with better accuracy-to-model size trade-off within frozen networks than within dense or randomly pruned source networks. In particular, freezing 70% of a ResNet on ImageNet provides 3.3× compression compared to the SLT found within a dense counterpart, raises accuracy by up to 14.12 points compared to the SLT found within a randomly pruned counterpart, and offers a better accuracy-model size trade-off than both.

## 1 Introduction

The strong lottery ticket hypothesis (SLTH) conjectured the existence of subnetworks within a randomly weighted network—*strong lottery tickets* (SLTs)—that achieve comparable accuracy to trained dense networks (Zhou et al., 2019; Ramanujan et al., 2020; Malach et al., 2020). The existence of such subnetworks that do not require weight training, illustrated in Figure 1 (left), has been demonstrated experimentally (Zhou

---

[*]Equal contribution. [†] Correspondence to: Hikari Otsuka <otsuka.hikari@artic.iir.isct.ac.jp>.

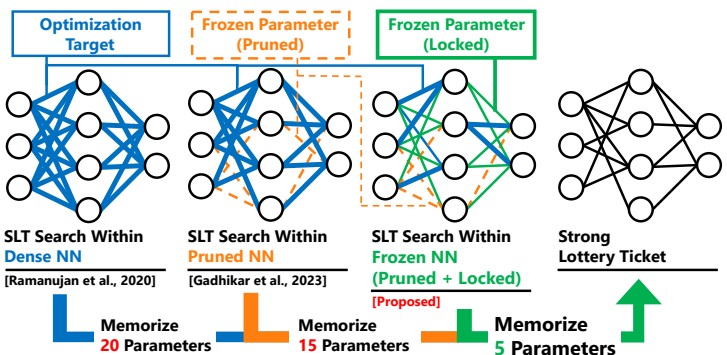

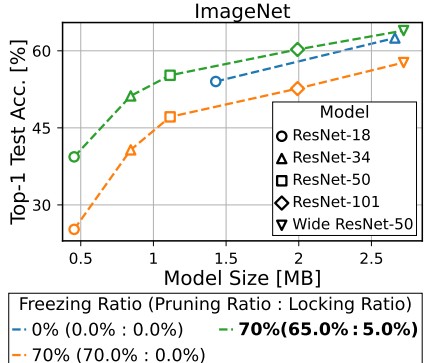

Figure 1: Freezing the source network by randomly pruning some parameters and locking others reduces the memorized supermask for finding an SLT.

Figure 2: Freezing (●) improves the accuracy-to-model size trade-off over pre-pruning only (●) or non-freezing (●).

et al., 2019; Ramanujan et al., 2020; Huang et al., 2022; López García-Arias et al., 2023; Yeo et al., 2023) and proven theoretically for dense source networks (Malach et al., 2020; Orseau et al., 2020; Pensia et al., 2020; Diffenderfer & Kailkhura, 2021; da Cunha et al., 2021; Burkholz, 2022a; Burkholz et al., 2022; Ferbach et al., 2023; Gadhikar et al., 2023). They appear in networks with excessive amounts of parameters, and their existence indicates the possibility of new optimization strategies for deep neural networks based on a network connectivity perspective.

SLTs offer a particularly advantageous opportunity for specialized inference hardware (Hirose et al., 2022; Chen et al., 2022; 2023) since the random weights can be reconstructed from the seed (i.e., they do not need to be memorized), and the binary mask—*supermask*—can be greatly compressed with entropy coding, vastly reducing off-chip memory access and its associated overheads. Furthermore, the binary nature of both the supermask and the random weights can be exploited for multiplier-less execution, as demonstrated practically by Hirose et al. (2022).

Recently, Gadhikar et al. (2023) showed that accurate SLTs could be found even if the number of edges to be optimized was reduced by randomly pruning the source network at initialization (see Figure 1, center). Since the random pre-pruning mask can be reconstructed from the seed in the same way as the random weights, and thus there is no need to store the part of the supermask corresponding to pre-pruned weights, their approach can be exploited to further reduce the memory cost required by the SLT in specialized hardware. However, when aiming for high compression, their method can only search for SLTs in the relatively high sparsity region, and may even lead to layer collapse (Hayou et al., 2020; Tanaka et al., 2020). Furthermore, since the SLT sparsity regions where highly accurate SLTs exist depend on the dataset and network architecture, a search limited to sparse regions may fail to find accurate SLTs. For instance, work on SLTs within graph neural networks has shown that dense SLTs are more accurate than sparse ones in some settings (Huang et al., 2022; Yan et al., 2023). Therefore, it would be desirable to use a method that allows for increased randomness in SLTs for further model compression, but has the freedom to allocate it in the SLT sparsity regions that lead to more accurate tickets.

This paper introduces such a novel method to reduce the memory cost of the optimized supermask without restricting the desired sparsity of SLTs to be searched for: in addition to random pruning at initialization, it also locks randomly chosen parameters at initialization to be a permanent part of the SLT (i.e., never pruned), as exemplified in Figure 1 (right). Both the randomly pruned and the locked parameters—the *frozen* parameters—are left completely random and can be regenerated from seed. The weights corresponding to the optimized supermask region (less than 50% of the total) are also reconstructed from seed, and the supermask is binary and sparse, producing a highly compressible model. Far from negatively impacting performance, this cost reduction is efficient: as shown in Figure 2, the frozen SLTs achieve a higher accuracy than SLTs with a similar size resulting from conventional methods. The contributions of this paper are summarized as follows:

- We propose a novel method that vastly reduces the number of parameters to be memorized for finding an accurate SLT by freezing (pruning and locking) the source random network.

- We experimentally validate our method in three scenarios corresponding to low, medium, and high optimal SLT sparsity regions, which reveal that parameter freezing consistently produces smaller yet accurate supermasks. Even with randomly frozen parameters, we find highly accurate SLTs that cannot be found within dense networks for some desired sparsities.

- Furthermore, the experimental results show that SLTs found in frozen networks achieve comparable or better accuracy-to-model size trade-off than SLTs found within dense (non-freezing) or sparse (non-locking) random networks.

As mentioned above, SLTs are quite attractive for neural engine design, as they can vastly reduce the memory size for model storage, meaning that off-chip memory access—by far the major bottleneck of energy and time consumption (Horowitz, 2014)—can be drastically reduced for energy-efficient inference acceleration (Hirose et al., 2022). Our contributions have the potential to reduce off-chip memory access further and to make inference more energy-efficient than previous designs.

## 2 Preliminaries

This section outlines the background of strong lottery tickets (SLTs) within dense or sparse networks and SLT search algorithms.

### 2.1 Strong Lottery Tickets in Dense Networks

SLTs (Zhou et al., 2019; Ramanujan et al., 2020; Malach et al., 2020) are subnetworks within a randomly weighted neural network that achieve high accuracy *without any weight training*. Compared with learned dense weight models, SLTs can be reconstructed from a small amount of information: since the random weights can be regenerated from their seed, it is only necessary to store the binary supermask and the seed (Hirose et al., 2022). SLT search algorithms for deep neural networks (Zhou et al., 2019; Ramanujan et al., 2020; Zhou et al., 2021; Sreenivasan et al., 2022) find SLTs by updating weight scores, which are then used to generate the supermask, instead of updating weights. For example, the `Edge-Popup` algorithm (Ramanujan et al., 2020; Sreenivasan et al., 2022) used in this paper finds SLTs by applying a supermask generated from the connections with the top-$k\%$ scores.

**SLT Existence via Subset-Sum Approximation**   Given a set of random variables and a target value, the subset-sum problem consists of finding a subset whose total value approximates the target. Lueker (1998) showed that such a subset exists with high probability if the number of random variables is sufficiently large:

**Lemma 2.1** (Subset-Sum Approximation (Lueker, 1998))**.** *Let $X_1, ..., X_n \sim U(-1, 1)$ be independent, uniformly distributed random variables. Then, except with exponentially small probability, any $z \in [-1, 1]$ can be approximated by a subset-sum of $X_i$ if $n$ is sufficiently large.*

Based on this Lemma 2.1, which was first introduced into the SLT context by Pensia et al. (2020), previous works (Burkholz, 2022a;b; da Cunha et al., 2021; Pensia et al., 2020) showed that an SLT that approximates an arbitrary target network exists in a dense source network if it is logarithmically wider and constantly deeper than the target network. In particular, Burkholz (2022a) proved that a source network with depth $L+1$ and larger width than the target network contains an SLT that can approximate the target network with depth $L$.

### 2.2 Strong Lottery Tickets in Sparse Networks

Recently, Gadhikar et al. (2023) revealed that SLTs also exist within sparse source networks, i.e., random networks that have been *randomly pruned at initialization* (see Figure 1, center). They showed its existence experimentally as well as theoretically, as outlined later: an SLT that approximates a given target network exists with high probability in a sparse source network that is sufficiently wider and deeper than the target.

**SLT Existence in Sparse Networks**   To prove the existence of SLTs within sparse networks, Gadhikar et al. (2023) extended the subset-sum approximation (Lemma 2.1) to the situation where randomly chosen variables are permanently pruned at initialization:

**Lemma 2.2** (Subset-Sum Approximation in Sparse Networks (Gadhikar et al., 2023)). *Let $X_1, ..., X_n$ be as in Lemma 2.1, and $M_1, ..., M_n \sim Ber(p)$ be independent, Bernoulli distributed random variables with $p \in (0, 1)$. Then, except with exponentially small probability, any $z \in [-1, 1]$ can be approximated by a subset-sum of $M_i X_i$ if $n$ is sufficiently large.*

By applying this extended lemma instead of Lemma 2.1 to the SLT existence theorem presented by Burkholz (2022a), Gadhikar et al. (2023) proved the SLT existence in sparse networks as following theorem:

**Theorem 2.1** (SLT Existence in Sparse Networks Gadhikar et al. (2023)). *Let a target network $f_T$ with depth $L$ and a sparse source network $f_S$ with depth $L+1$ be given. Assume that the source network is randomly pruned with pruning ratio $p_l$ for each $l$-th layer. Also assume that these networks have ReLU activation function and initialized by using a uniform distribution $U[-1, 1]$. Then, except with exponentially small probability, a subnetwork $f_{SLT}$ exists in the sparse source network $f_S$ such that $f_{SLT}$ approximates the target network $f_T$ if the width of $f_S$ is sufficiently large.*

Thus, by extending the subset-sum approximation lemma, the conventional SLT existence proof can be easily extended to various network settings.

## 3   Strong Lottery Tickets in Frozen Networks

The approach of finding SLTs within sparse networks by Gadhikar et al. (2023) offers a practical advantage: as the randomly pre-pruned parts can be reconstructed from the seed, this pre-pruning process can be exploited to reduce the memory requirement of the supermask.

However, this approach also imposes the limitation that it can only search for SLTs with a sparsity higher than the pre-pruning ratio. For example, a random pre-pruning ratio of 90% vastly reduces the memory size of the supermask, but it limits the search to only SLTs sparser than 90%. Therefore, their method is incompatible with exploring the whole SLT sparsity range for optimal accuracy.

This paper proposes a novel method that allows us to find SLTs within the optimal sparsity range while still capitalizing on the compression gains offered by random connectivity initialization. We explore the performance of SLTs within a *frozen source network*, i.e., a random network that (in addition to randomly pruned parameters) has randomly chosen parameters forced to be permanent part (i.e., never pruned) of the SLT—*locked parameters* (see Figure 1, right). Since the random locking pattern can be reconstructed from seed in the same way as the random pre-pruning pattern, our freezing method allows us to compress SLTs further. Additionally, locking allows us to extend the benefits of pre-pruning to the scenarios where the SLT sparsity should not be too high, e.g., as found in graph neural networks (Huang et al., 2022; Yan et al., 2023).

This section first describes the construction of frozen networks and the freezing pattern encoding for model compression. Then, we perform a preliminary experiment of the optimal settings for the proposed method in preparation for the evaluation experiments in Section 4.

### 3.1   Partial Freezing for Enhanced SLT Compression

In our method, since frozen regions in the source network are completely random and thus can be reconstructed from seed, we need to memorize only the optimized supermask region. The total amount of frozen parameters is determined by a global *freezing ratio* $F_r$, which is the sum of the respective *pre-pruning ratio* $P_r$ and *locking ratio* $L_r$. Therefore, we optimize the $1 - F_r$ of the parameters (i.e., non-frozen parameters) of the original dense network. As explained previously, it is not possible to search for an SLT with a lower sparsity (denser) than the frozen source network. Consequently, as shown in Figure 3, the pre-pruning ratio $P_r$ sets the lower bound of the sparsity of the SLTs that can be found. On the other hand, it is not possible to search for an SLT that prunes more parameters than those available, so the locking ratio $L_r$ sets the upper bound of the sparsity of the SLTs that can be found. (Specifically, the upper bound is given by $1 - L_r$.) Thus, these ratios

allow us to freely control the proportion of pre-pruning and locking ratios and memory size reduction ratio for the desired SLT sparsity.

**Frozen network construction:** We assume that the $l$-th layer of network with depth $L$ has weights $\boldsymbol{w}^{(l)} \in \mathbb{R}^{n^{(l)}}$. The number of frozen parameters is determined by the layer-wise *freezing ratios* $p_f^{(1)}, ..., p_f^{(L)}$, where each $p_f^{(l)}$ is defined as the sum of the layer-wise *pre-pruning ratio* $p_p^{(l)}$ and the layer-wise *locking ratio* $p_l^{(l)}$, and $p_f^{(l)}, p_p^{(l)}, p_l^{(l)} \in [0,1]$.

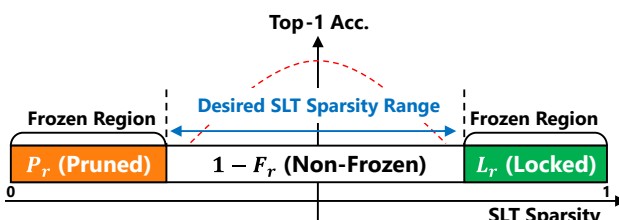

Figure 3: Pre-pruning and locking set the bounds of the SLT sparsity that can be found. These optimal bounds are investigated in Section 3.2.

To prune $p_f^{(l)} n^{(l)}$ parameters and lock $p_l^{(l)} n^{(l)}$ parameters, we generate two random masks, a pre-pruning mask $\boldsymbol{m}_p^{(l)} \in \{0,1\}^{n^{(l)}}$ and a locking mask $\boldsymbol{m}_l^{(l)} \in \{0,1\}^{n^{(l)}}$, so that they satisfy $||1 - \boldsymbol{m}_p^{(l)}|| = p_p^{(l)} n^{(l)}$, $||\boldsymbol{m}_l^{(l)}|| = p_l^{(l)} n^{(l)}$, and $(1 - \boldsymbol{m}_p^{(l)}) \cdot \boldsymbol{m}_l^{(l)} = 0$ (i.e., we require pre-pruning and locking to be implemented without overlap). The layer-wise weights frozen with these masks are calculated as $\boldsymbol{w}_f^{(l)} := (\boldsymbol{m}_p^{(l)} \odot (1 - \boldsymbol{m}_l^{(l)}) + \boldsymbol{m}_l^{(l)}) \odot \boldsymbol{w}^{(l)}$. These masks are fixed during training, and we search for SLTs only in the parts where $\boldsymbol{m}_p^{(l)} \odot (1 - \boldsymbol{m}_l^{(l)})$ is one.

**Setting the layer-wise ratios:** Our method considers the following two existing strategies for determining the layer-wise pre-pruning ratio from the desired global pre-pruning ratio of the network:

- *Erdős-Rényi-Kernel* (ERK): The pre-pruning ratio of layer $l$ is proportional to the scale $(C_{in}^{(l)} + C_{out}^{(l)} + k_h^{(l)} + k_w^{(l)})/(C_{in}^{(l)} \cdot C_{out}^{(l)} \cdot k_h^{(l)} \cdot k_w^{(l)})$, where $C_{in}^{(l)}$, $C_{out}^{(l)}$, $k_h^{(l)}$, and $k_w^{(l)}$ denote input channels, output channels, kernel height, and kernel width of the layer $l$, respectively (Evci et al., 2020).

- *Edge Per Layer* (EPL): Each layer's pre-pruning ratio is set so that they all have the same number of remaining weights (Price & Tanner, 2021; Gadhikar et al., 2023).

The same strategy is employed to determine the layer-wise pre-pruning and freezing ratios from their respective global ratios, and then the locking ratios are calculated as the difference between the corresponding freezing and pre-pruning ratios.

**Freezing pattern encoding for model compression:** The freezing pattern can be encoded during inference as a ternary mask—a *freezing mask*—that indicates whether a parameter is pruned, locked, or part of the supermask. For example, encoding pre-pruning as $-1$, locking as $+1$, and supermask inclusion as $0$, the layer-wise freezing mask can be encoded as $\boldsymbol{m}_l^{(l)} + (\boldsymbol{m}_p^{(l)} - 1) \in \{-1, 0, 1\}^{n^{(l)}}$. Since this freezing mask is also random and fixed, it can be regenerated from its seed and ratios, similarly to the random weights. Furthermore, the supermask size is reduced by excluding from it the frozen parameters, so the SLTs found by this method can be compressed in inference to an even smaller memory size than those produced by the existing SLT literature (Hirose et al., 2022; Okoshi et al., 2022; López García-Arias et al., 2023), reducing costly off-chip memory access on specialized neural inference accelerators (Hirose et al., 2022; Chen et al., 2023), and thus offering an opportunity to perform even faster and more energy-efficient inference processing.

## 3.2 Optimal Pruning:Locking Proportion for Freezing

Here, we perform a preliminary investigation of the optimal pruning:locking proportion for each given desired SLT sparsity by varying the proportion with a fixed freezing ratio of the network. Figure 4 explores different configurations of a 80% freezing ratio—the situation where the supermask memory size is 20% of the SLT in the dense source network—on a Conv6 network and compares the performance of the found SLTs on

CIFAR-10. As expected, the best-performing SLT of each prune:lock configuration is found at the center of the non-frozen region, where the number of candidate subnetworks is maximized. In other words, when the non-frozen region accounts for $S \in [0, 1]$ of the entire source network, the optimal SLT sparsity is $k = P_r + S/2$, where $P_r = \sum_l p_p^{(l)} n^{(l)} / \sum_k n^{(k)}$ is the global pre-pruning ratio of the network. Conversely, for a given freezing ratio $F_r$ of the network and a desired SLT sparsity $k$, the optimal position of the frozen region is set by the pre-pruning ratio $P_r = k - (1 - F_r)/2$ and the corresponding locking ratio $L_r = F_r - P_r$. In the cases where this would result in $P_r < 0$ or $L_r < 0$, we choose a best-effort approach that keeps the desired freezing ratio and sets the bounds to $P_r = 0$ or $L_r = 0$, respectively.

Additionally, Figure 4 compares the two strategies for setting ratios considered in Section 3.1, showing that EPL outperforms ERK in all cases. Consequently, hereafter, the proposed method sets the global ratios in order to position the frozen region center as close as possible to the desired SLT sparsity, and then sets the layer-wise ratios using EPL.

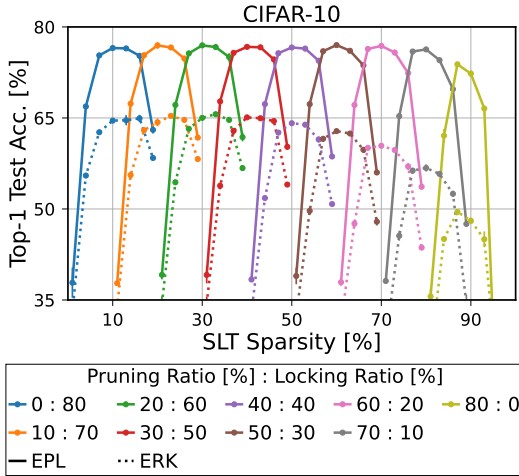

Figure 4: Different prune:lock proportions of a 80% freezing ratio using a Conv6.

### 3.3 SLT Existence in Frozen Networks

One question comes to mind here: *does SLT exist that approximates a given target network, even if the parameters are randomly frozen (i.e., pruned or locked) in advance?* It has been shown by Gadhikar et al. (2023) that an SLT that approximates a given target network exists in the pre-pruned source network if the source network is sufficiently wider and deeper than the target, but it is not known whether such an SLT exists in the situation of a frozen source network. Here, we provide a rough theoretical result indicating that an SLT capable of approximating a target network exists in a frozen network with a sufficiently large width. This result is proved by extending the subset-sum approximation lemma (Lemma 2.1) to the case where some parameters are locked (for detailed proof, see Appendix A).

**Lemma 3.1** (Subset-Sum Approximation in Randomly Locked Networks)**.** *Let $X_1, ..., X_n$ be as in Lemma 2.1, and $M'_1, ..., M'_n \sim Ber(q)$ be independent, Bernoulli distributed random variables with $q \in (0, 1)$. Then, except with exponentially small probability, any $z \in [-1, 1]$ can be approximated by the sum of $\sum_{i=1}^{n} M'_i X_i$ and a subset-sum of $(1 - M'_i)X_i$ if $n$ is sufficiently large.*

Then, by using the same proof procedure as Lemma 3.1, we extend the subset-sum approximation to the situation where some random variables are frozen (i.e., pruned or locked).

**Lemma 3.2** (Subset-Sum Approximation in Frozen Networks)**.** *Let $X_1, ..., X_n$ be as in Lemma 2.1, $M_1, ..., M_n \sim Ber(p)$ be as in Lemma 2.2, and $M'_1, ..., M'_n \sim Ber(q)$ be as in Lemma 3.1. Then, except with exponentially small probability, any $z \in [-1, 1]$ can be approximated by the sum of $\sum_{i=1}^{n} M_i M'_i X_i$ and a subset-sum of $M_i(1 - M'_i)X_i$ if $n$ is sufficiently large.*

Finally, by applying Lemma 3.2 to the Theorem 2.5 in Gadhikar et al. (2023) instead of Lemma 2.2, it follows that an SLT approximating a target network exists within a frozen network.

**Theorem 3.1** (SLT Existence in Frozen Networks)**.** *Let a target network $f_T$ with depth $L$ and a partially frozen source network $f_S$ with depth $L+1$ be given. Assume that the source network is randomly frozen with pruning ratio $p_l$ and locking ratio $q_l$ for each $l$-th layer, except for $0$-th and $1$-st layers. Also assume that these networks use the ReLU activation function and are initialized with a uniform distribution $U[-1, 1]$. Then, except with exponentially small probability, a subnetwork $f_{SLT}$ exists in the frozen source network $f_S$ such that $f_{SLT}$ approximates the target network $f_T$ if the width of $f_S$ is sufficiently large.*

Note that this theorem is intended to provide an existential guarantee for the question raised at the beginning of this subsection, whether the freezing operation hurts the existence of SLTs that can approximate an arbitrary network; thus, we do not focus on the sharpness of the probability bound to keep the results simple.

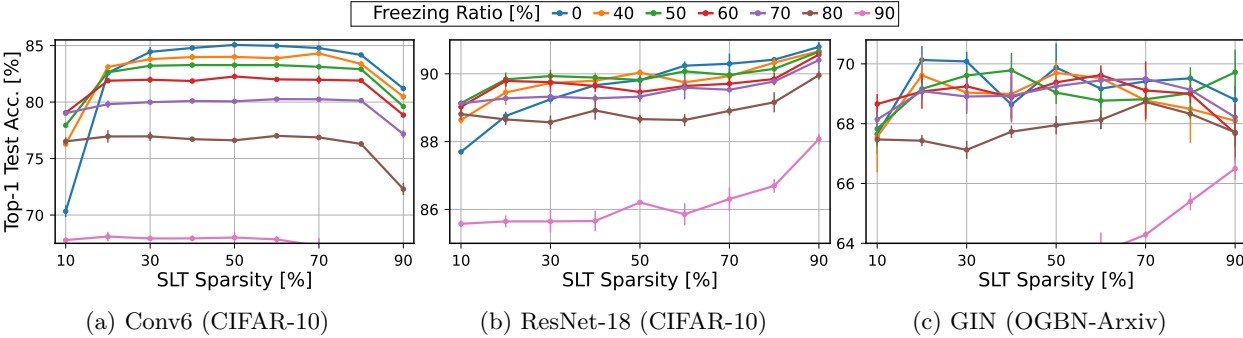

Figure 5: Impact of the freezing ratio on different architectures. Pruning and locking ratios are set following Section 3.2.

Nevertheless, proving sharper bounds may lead to capturing the effect of the freezing operation, which is left for future work.

## 4 Experiments

This section demonstrates that freezing reduces the SLT memory size in a broad range of situations by evaluating it on image classification and graph node classification. We evaluate the impact of the freezing ratio on various network architectures, identifying three scenarios. Then, we explore trade-offs between accuracy and model memory size for different network widths and architectures.

### 4.1 Experimental Settings

We evaluate the SLTs within frozen networks on image classification using the CIFAR-10 (Krizhevsky, 2009) and ImageNet (Russakovsky et al., 2015) datasets, and on node classification using the OGBN-Arxiv (Hu et al., 2020) dataset. CIFAR-10 and ImageNet train data are split into training and validation sets with a 4:1 ratio, while for OGBN-Arxiv we use the default set split. We test the models with the best validation accuracy and report the mean of three experiment repetitions for CIFAR-10 and OGBN-Arxiv, and the result of one experiment for ImageNet. The standard deviation of experiments conducted more than once is plotted as error bars.

For image classification we employ the VGG-like Conv6 (Simonyan & Zisserman, 2014; Ramanujan et al., 2020), ResNet (He et al., 2016), and Wide ResNet (Zagoruyko & Komodakis, 2016) architectures, and for graph node classification the 4-layer modified GIN (Xu et al., 2019) architecture in Huang et al. (2022), all implemented with no learned biases. ResNet and GIN use non-affine Batch Normalization (Ioffe & Szegedy, 2015), while Conv6 has no normalization layers. Random weights are initialized with the Kaiming Uniform distribution, while weight scores are initialized with the Kaiming Normal distribution (He et al., 2015).

SLTs are searched using an extension of `Edge-Popup` (Ramanujan et al., 2020) that enforces the desired SLT sparsity globally instead of per-layer (Sreenivasan et al., 2022). On CIFAR-10, scores are optimized for 100 epochs using stochastic gradient descent with momentum 0.9, batch size 128, weight decay 0.0001, and initial learning rates of 0.01 and 0.1 for Conv6 and ResNet-18, respectively. On ImageNet, scores are optimized by the same setting as ResNet-18 on CIFAR-10, but a 256 batch size. On OBGN-Arxiv, scores are optimized for 400 epochs using AdamW (Loshchilov & Hutter, 2019) with weight decay 0.0001 and initial learning rate 0.01. All experiments use cosine learning rate decay (Loshchilov & Hutter, 2017). These can be adequately verified with two NVIDIA H100 SXM5 94GB GPUs.

### 4.2 Varying Desired SLT Sparsity at Different Freezing Ratios

This section investigates the effect of the freezing ratio of the network on desired SLT accuracy. We identify three scenarios, represented in Figure 5.

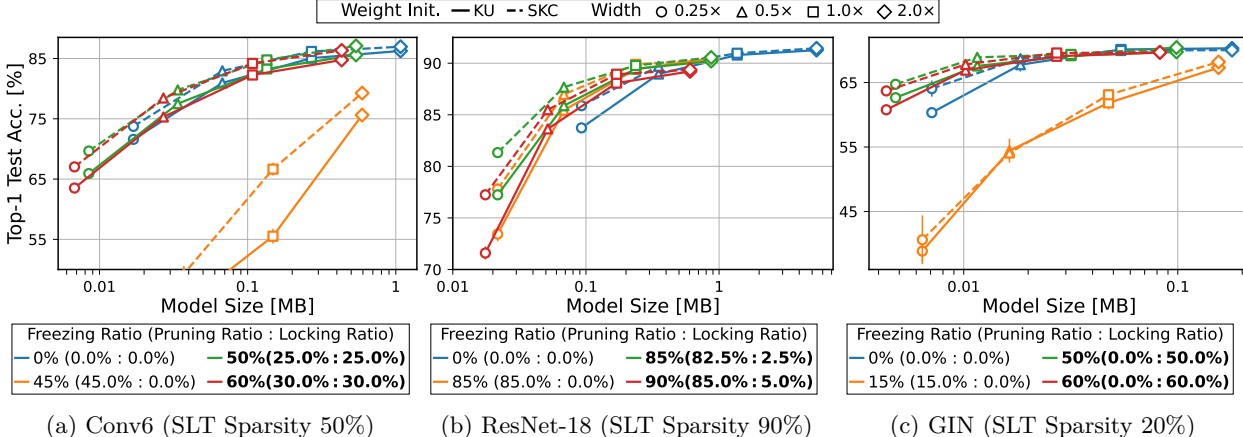

(a) Conv6 (SLT Sparsity 50%)          (b) ResNet-18 (SLT Sparsity 90%)          (c) GIN (SLT Sparsity 20%)

Figure 6: Compared to sparse (●) or dense (●) source networks, freezing achieves better accuracy-to-model memory size trade-off (top-left is better).

In the cases where the optimal SLT sparsity is found at intermediate sparsity—e.g., around 50% for Conv6 in Figure 5a—pruning and locking can be applied with equally high ratios. Results show that the supermask memory size can be reduced by 40% with a small impact on accuracy, and by 70% with still a moderate accuracy drop of 5 points.

Applying the much larger ResNet-18 to the same task results in much stronger overparametrization. Therefore, optimal SLTs are found in the higher sparsity range, as revealed by Figure 5b, benefiting from much higher pruning than locking. Even though 90% of the memory size is reduced in this scenario, we can find 90% sparse SLTs with 88.1% accuracy.

As an example of the scenario where optimal SLT sparsities are found in the denser range, benefiting from a higher locking ratio, we evaluate GIN in Figure 5c. Compared with the best-performing SLT found in the dense GIN, of 70.1% accuracy and 20% sparsity, by freezing 50% of the memory size, our method finds similarly performing SLTs of 69.8% accuracy with 40% sparsity.

Interestingly, with low SLT sparsity (e.g., 10% sparsity) in all three scenarios, despite the reduced parameters to be optimized, SLTs within frozen networks achieve higher accuracy than SLTs within dense networks. These results imply that parameter freezing at an appropriate ratio has the effect of avoiding the inclusion of low-grade local optimal solutions in the search space. While searching for SLTs in a dense network by `Edge-Popup` leads to convergence to a local optimal solution (Fischer & Burkholz, 2022), a moderate random parameter freezing may reduce the number of less accurate local optimal solutions and facilitate convergence to a more accurate local optimal solution within the reduced search space. In other words, we conjecture that if the network is properly frozen, the local optimal solution for the reduced search space is close to the global optimal solution of the entire search space.

### 4.3 Accuracy-to-Model Memory Size Trade-Off

The freezing mask compression scheme proposed in Section 3.1 allows to reduce the model memory size during inference by regenerating both the random weights and the freezing mask with random number generators. Here we consider this compression and investigate the accuracy-to-model size trade-off offered by the SLTs found in the frozen networks. Model size refers to the total memory size of model parameters that need to be stored. The random weights and the frozen parts of the supermask are excluded, since they can be regenerated from seed, whereas each non-frozen element of the supermask and each learned batchnorm parameter are counted as 1 and 32 bits, respectively. Furthermore, we also compare the Kaiming Uniform (KU) weight initialization used so far with the binary weights provided by the Signed Kaiming Constant (SKC) initialization (Ramanujan et al., 2020), which can be exploited for reduced computational cost in neural engines (Hirose et al., 2022). For SKC, we scale weights by $1/\sqrt{1-k_l}$, where $k_l$ is the sparsity of each

layer, as proposed by Ramanujan et al. (2020). Note that since we are interested in the accuracy of the SLTs compressed as much as possible obtained by each method, the pre-pruning ratio for the pruning-only case is determined as (target sparsity $- 5$)% to compress the SLT as much as possible while allowing to search for SLTs. On the other hand, we evaluate freezing with a higher compression ratio than pruning only. This setting means we evaluate our method under more strict conditions than pruning-only from the compression ratio perspective.

Figure 6 explores varying the width of the source network to analyze its impact on accuracy and model size. SLT sparsity is fixed to that of the best performing SLT found in a dense source network in Figure 5: 50% in Conv6, 90% in ResNet-18, and 20% in GIN. Compared to the SLTs found in dense or sparse source networks, SLTs found by our method achieve similar or higher accuracy for similar or smaller model size, thus improving the accuracy-to-model size trade-off in all scenarios.

Empirically, it is known that SLTs within an SKC-initialized dense network achieve better performance than with continuous random weights (Ramanujan et al., 2020; Okoshi et al., 2022; López García-Arias et al., 2023; Yan et al., 2023). Our results show that such a trend can also be observed in frozen source networks. Nonetheless, in all source networks, we find that this improvement is smaller the wider the source network is, suggesting that the requirement for source networks of larger width is weaker in the case of binary weights.

### 4.4 ImageNet Experiments

Finally, we evaluate our method using larger models on a large-scale dataset: deeper and wider ResNets on ImageNet. Since SLTs with 80% sparsity achieve the highest accuracy in a dense source ResNet-50 (for details, see Appendix B.1), we compare the three methods using 80% SLT sparsity.

Figure 2 compares the accuracy-to-model size trade-off in finding the 80% sparsity SLTs between the proposed and conventional methods with SKC using ImageNet. Despite the more challenging setting and the significant 70% memory size reduction, our method (green) finds SLTs that are more accurate than pre-pruning-only methods (orange) for the same model size. This result demonstrates that the effective combination of parameter pruning and locking at initialization can improve the SLT memory efficiency even on large-scale datasets and models.

### 4.5 Result Analysis

Table 1 summarizes the presented results and compares the proposed method to weight training and SLT search in dense and sparse source networks for the same network architecture and SLT sparsity. As described in Section 4.3, the model size is calculated as the sum of the unfrozen partial supermask (1 bit/parameter) and the batch normalization parameters (32 bits/parameter). Also, as described in Section 4.2, the target SLT sparsity is fixed to that of the best-performing SLT found in a dense source network, and we determine the pre-pruning ratio in sparse network as (SLT sparsity $- 5$)% for high compression.

In the case of Conv6, compared to weight training our method provides reductions of 66.4× of the model size, in exchange of a small accuracy drop. Compared to the SLT found in the sparse source network, the SLT found in the frozen source network achieves much higher accuracy.

Even in the more challenging case of ResNet-18, SLTs found in a frozen network are 250.8× smaller than the trained-weight model. When comparing SLTs found in sparse and frozen networks with the same 85% freezing ratio, the accuracy is almost equivalent, demonstrating that the inclusion of locking does not introduce a degradation in accuracy even in the scenario that benefits more from pruning.

Interestingly, the SLT found in the frozen GIN model achieves comparable accuracy to the trained-weight network, even though the model size is reduced by 40.3×. The SLT found within a dense network also achieves accuracy comparable to that of the trained-weight network, but the SLT found by our method is smaller. Generally, graph neural networks (GNNs) suffer from a generalization performance degradation due to the over-smoothing problem Li et al. (2018). Searching for SLTs within dense GNNs has been shown to mitigate the over-smoothing problem Huang et al. (2022) and achieve higher accuracy. As our method is

Table 1: Comparison between trained-weight networks and SLTs found in dense, sparse, and frozen networks.

| Method (Source Net.) | Weight Init. | Sparsity [%] | Pruning Ratio [%] | Locking Ratio [%] | Top-1 Test Acc. [%] | Model Size [MB] |
|---|---|---|---|---|---|---|
| Conv6 & CIFAR-10 | | | | | | |
| Weight Training | KU | - | - | - | 87.1 | 8.63 |
| SLT (Dense) | SKC | 50 | 0 | 0 | 86.2 | 0.27 |
| SLT (Sparse) | SKC | 50 | 45 | 0 | 66.7 | 0.15 |
| **SLT (Frozen)** | SKC | 50 | 25 | 25 | **84.8** | **0.13** |
| ResNet-18 & CIFAR-10 | | | | | | |
| Weight Training | KU | - | - | - | 92.4 | 42.63 |
| SLT (Dense) | SKC | 90 | 0 | 0 | 91.0 | 1.37 |
| SLT (Sparse) | SKC | 90 | 85 | 0 | 89.9 | 0.24 |
| **SLT (Frozen)** | SKC | 90 | 82.5 | 2.5 | **89.8** | 0.24 |
| **SLT (Frozen)** | SKC | 90 | 85 | 5 | **88.9** | **0.17** |
| GIN & OGBN-Arxiv | | | | | | |
| Weight Training | KU | - | - | - | 70.1 | 1.452 |
| SLT (Dense) | SKC | 20 | 0 | 0 | 70.0 | 0.054 |
| SLT (Sparse) | SKC | 20 | 15 | 0 | 62.3 | 0.047 |
| **SLT (Frozen)** | SKC | 20 | 0 | 40 | **69.2** | **0.036** |
| ResNet-50 & ImageNet | | | | | | |
| Weight Training | KU | - | - | - | 74.4 | 97.49 |
| SLT (Dense) | SKC | 80 | 0 | 0 | 66.8 | 3.24 |
| SLT (Sparse) | SKC | 80 | 70 | 0 | 47.1 | 1.11 |
| **SLT (Frozen)** | SKC | 80 | 65 | 5 | **55.2** | **1.11** |
| ResNet-34 & ImageNet | | | | | | |
| SLT (Dense) | SKC | 80 | 0 | 0 | 62.5 | 2.66 |
| SLT (Sparse) | SKC | 80 | 70 | 0 | 40.7 | 0.84 |
| **SLT (Frozen)** | SKC | 80 | 65 | 5 | **51.2** | **0.84** |
| ResNet-18 & ImageNet | | | | | | |
| SLT (Dense) | SKC | 80 | 0 | 0 | 54.0 | 1.43 |
| SLT (Sparse) | SKC | 80 | 70 | 0 | 25.2 | 0.45 |
| **SLT (Frozen)** | SKC | 80 | 65 | 5 | **39.4** | **0.45** |
| Wide ResNet-50 & ImageNet | | | | | | |
| SLT (Dense) | SKC | 80 | 0 | 0 | 70.8 | 8.46 |
| SLT (Sparse) | SKC | 80 | 70 | 0 | 57.7 | 2.72 |
| **SLT (Frozen)** | SKC | 80 | 65 | 5 | **63.9** | **2.72** |

similarly accurate compared to dense networks, it is possible that parameter freezing offers an even stronger solution to the over-smoothing problem.

Although the ImageNet experiment falls into the scenario where a relatively sparse SLT is more accurate (like the ResNet-18 experiment on CIFAR-10), the SLT found in a sparse network is significantly less accurate. On the other hand, the inclusion of parameter locking shows a significant improvement in SLT accuracy. These results suggest that the number of highly accurate SLT patterns decreases as the difficulty of the problem increases, and the pruning:locking proportion affects the performance of the found SLTs more severely.

## 5 Limitations

Our results have the following limitations: 1) Although the proposed method can effectively reduce the model memory size for inference in principle, actual hardware implementation remains for future work. The previous work of the SLT-specialized neural engine, Hiddenite Hirose et al. (2022), would be a promising direction for such future implementation. 2) This paper provides theoretical support for the existence of SLTs in frozen networks, but it does not provide their superiority in the approximation capability compared to random pruning, a special case of random freezing, which may depend on the sparsity nature of target networks. Also, the theoretical consideration of the appropriate proportion of these ratios is left for future work. 3) Even though we demonstrate our method on various model architectures including CNNs, ResNet families and GIN in Section 4, following previous SLT work (Zhou et al., 2019; Ramanujan et al., 2020; López García-Arias et al., 2023; Huang et al., 2022; Yan et al., 2023), we have to leave it for future work to apply our method to Transformers (Vaswani et al., 2017) because SLT itself has not yet been established for Transformers.

## 6 Conclusion

This paper capitalizes on the fact that pre-pruning a randomly weighted network reduces the supermask memory size, but identifies that doing so limits the search to a suboptimal sparsity region. This problem is tackled by freezing (i.e., pruning or locking) some parameters at initialization, excluding them from the search. Freezing allows to the search for SLTs in the optimal sparsity region, while further reducing the model size. Experimental results show that SLTs found in frozen networks improve the accuracy-to-model size trade-off compared to SLTs found in dense (Ramanujan et al., 2020; Sreenivasan et al., 2022) or sparse networks (Gadhikar et al., 2023). Interestingly, although for weight training random parameter locking has only been found useful for reducing accuracy degradation in network compression (Zhou et al., 2019; Wimmer et al., 2020), we identify scenarios in SLT training where they can be used for raising accuracy. Our method can be interpreted as being capable of generating more useful SLT information from a random seed than previous methods, offering an opportunity for reducing off-chip memory access in specialized SLT accelerators (Hirose et al., 2022). Additionally, the reduced number of parameters to be optimized may be exploited for training cost reduction. Appropriate freezing may improve the SLT search efficiency, including preventing convergence to local low-precision solutions as experimentally confirmed in Section 4.2, and improving convergence speed through reduced parameters to be optimized. Investigations about the SLT search efficiency remain as future work.

## Acknowledgments

This work was supported in part by JSPS KAKENHI Grant Number JP23H05489 and JST-ALCA-Next Japan Grant # JPMJAN24F3.

## Impact Statements

This paper presents work whose goal is to advance the field of machine learning. Among the many potential societal consequences of our work, we highlight its potential to reduce the computational cost of inference of neural networks, and thus their energy consumption footprint.

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

# A    Proof for Strong Lottery Tickets (SLTs) Existence in Frozen Networks

This section describes the proofs of the lemma and theorem introduced in the manuscript.

We first extend the lemma of the subset-sum approximation shown by Lueker (1998) to the case where some random variables are always included in the subset-sum.

**Lemma A.1** (Subset-Sum Approximation in Randomly Locked Networks). *Let $X_1, \cdots, X_n \sim U(-1, 1)$ be independent, uniformly distributed random variables, and $M_1, \cdots, M_n \sim \mathrm{Ber}(q)$ be independent, Bernoulli distributed random variables with $q \in (0, 1)$. Let $0 < \varepsilon, \delta < 1$. Then, with probability at least $1 - \delta$, for any $z \in [-1, 1]$, there exist indices $I \subset \{1, \cdots, n\}$ such that $|z - \sum_{i=1}^{n} M_i X_i - \sum_{i \in I} (1 - M_i) X_i| \leq \varepsilon$ if*

$$n \geq C \log\left(\frac{C'}{\varepsilon\delta}\right),\tag{1}$$

*where $C, C' > 0$ are constants, and $C$ depends on $q$.*

*Proof.* Let $M_1, \cdots, M_n \sim \mathrm{Ber}(q)$ and $m := \sum_i M_i$. By Hoeffding's inequality, we have

$$\mathbb{P}\left(|m - qn| \leq \varepsilon_M n\right) \geq 1 - 2\exp\left(-\varepsilon_M^2 n\right)\tag{2}$$

for $\varepsilon_M > 0$. Thus, if we set $n \geq \frac{\log(2/\delta)}{\varepsilon_M^2}$ and $\varepsilon_M := \frac{\min(q, 1-q)}{2}$, we have

$$(2q - \beta)n \leq m \leq \beta n\tag{3}$$

with $\beta := q + \varepsilon_M = \min(\frac{3q}{2}, \frac{1+q}{2}) \in (0, 1)$, with probability at least $1 - \delta$. In particular, the number of non-vanishing terms in the sum $\sum_{i \in I}(1 - M_i)X_i$ is $n - m \geq (1 - \beta)n$ as long as each $X_i$ is non-zero.

Now fix $M_1, \cdots, M_n \sim \mathrm{Ber}(q)$ with $(2q - \beta)n \leq m \leq \beta n$. The goal is to approximate $z \in [-1, 1]$ and $\sum_{i=1}^{n} M_i X_i$ by subset sum from $\{(1 - M_i)X_i : M_i = 0\}$. For simplicity, we split it into two parts:

$$\{(1 - M_i)X_i : M_i = 0\} = \{X_1', \cdots, X_{n_1}'\} \cup \{X_1'', \cdots, X_{n_2}''\},\tag{4}$$

where the former part is used for approximating $z \in [-1, 1]$ and the latter part for approximating $\sum_{i=1}^{n} M_i X_i$.

To approximate $z \in [-1, 1]$, we can directly apply Corollary 2.5 from Lueker (1998):

$$\mathbb{P}\left(\forall z \in [-1, 1], \exists I \subset \{1, \cdots, n_1\} \text{ s.t. } |z - \sum_{i \in I} X_i'| \leq \varepsilon\right) \geq 1 - \delta\tag{5}$$

whenever $n_1 \geq 4C \log(8/\varepsilon\delta)$.

To approximate $\sum_{i=1}^{n} M_i X_i$, we have to evaluate its norm. By Hoeffding's inequality on $X_i$'s with $M_i \neq 0$, we have

$$\mathbb{P}\left(\left|\sum_{i=1}^{n} M_i X_i\right| \leq \alpha m\right) \geq 1 - 2\exp\left(-\frac{3\alpha^2 m}{2}\right),\tag{6}$$

for any fixed $\alpha > 0$, whose value will be specified later. Thus $\left|\sum_{i=1}^{n} M_i X_i\right| \leq \alpha m \leq \alpha \beta n$ holds with probability at least $1 - \delta$ whenever $m \geq \frac{2\log(2/\delta)}{3\alpha^2}$. Since $m \leq \beta n$ holds, $n \geq \frac{2\log(2/\delta)}{3\alpha^2\beta}$ is enough.

From the proof of Corollary 3.1 in Lueker (1998), for any $\gamma \in (0, \frac{1}{4})$, we know that

$$\mathbb{P}\left(\forall z \in [-\gamma n_2, \gamma n_2], \exists I \subset \{1, \cdots, n_2\} \text{ s.t. } |z - \sum_{i \in I} X_i''| \leq \varepsilon\right)\tag{7}$$

$$\geq 1 - \delta - 2\exp\left(-\frac{(1 - 4\gamma)^2 n_2 - 4}{32}\right)\tag{8}$$

whenever $n_2 \geq 2C \log(\frac{2}{\varepsilon\delta})$. Thus if

$$n_2 \geq \max\left(2C \log\left(\frac{2}{\varepsilon\delta}\right), \frac{C'}{(1-4\gamma)^2} \log\left(\frac{C''}{\delta}\right)\right) \tag{9}$$

holds, we can approximate any $z \in [-\gamma n_2, \gamma n_2]$ by subset sum of $X_1'', \cdots, X_{n_2}''$ with probability $1 - 2\delta$.

Now we assume $(\alpha\beta/\gamma)n \leq n_2$. For example, if we set $\alpha = \frac{1-\beta}{10-\beta}$ and $\gamma = \frac{1}{5}$, the assumption is satisfied when we split $\{(1 - M_i)X_i\}$ by $n_1 = n_2 = \frac{(1-\beta)n}{2}$ in (4). Then the approximation (7) can be applied to $z = \sum_{i=1}^n M_i X_i$ with high probability since $|\sum_{i=1}^n M_i X_i| \leq \alpha m \leq \alpha\beta n \leq \gamma n_2$ holds.

By combining (5), (6), (9) with the fact that $n = m + n_1 + n_2$, and replacing $\varepsilon$ and $\delta$ with $\varepsilon/2$ and $\delta/6$, respectively, we obtain the desired results. $\qquad\square$

By similar arguments as in Lueker (1998) or Pensia et al. (2020), we can easily generalize Lemma A.1 to the case where the distribution followed by $X_i$ contains the uniform distribution. Also, by using the same proof procedure, we can prove the following extension of Lemma A.1:

**Lemma A.2** (Subset-Sum Approximation in Frozen Networks.). *Let $X_1, ..., X_n$ be independent, uniformly distributed random variables so that $X \sim U(-1, 1)$, $M_1, ..., M_n$ be independent, Bernoulli distributed random variables so that $M_i \sim Ber(p)$ for $p \in (0, 1)$, and $M_1', ..., M_n'$ be independent, Bernoulli distributed random variables so that $M_i' \sim Ber(q)$ for $q \in (0, 1)$. Let $\varepsilon, \delta \in (0, 1)$ be given. Then for any $z \in [-1, 1]$ there exists a subset $I \subseteq \{1, ..., n\}$ so that with probability at least $1 - \delta$ we have $\left| z - \sum_{i=1}^n M_i M_i' X_i - \sum_{i \in I} M_i (1 - M_i') X_i \right| \leq \varepsilon$ if*

$$n \geq C \log\left(\frac{C'}{\varepsilon\delta}\right), \tag{10}$$

*where $C, C' > 0$ are constants, and $C$ depends on $p$ and $q$.*

These corollaries seem to be derived by taking $z' := z - \sum_{i=1}^n M_i X_i$ as the target of the Lueker (1998) and Gadhikar et al. (2023) corollaries. However, it cannot. Their corollaries need the target to be deterministic and have constant bounds. On the other hand, since $\sum_{i=1}^n M_i X_i$ is stochastic, $z'$ is a stochastic value, and the bounds of $z'$ are not constant. Therefore, $z'$ cannot be directly assigned to the conventional corollaries, and our approach is necessary for proofing the subset-sum approximation with frozen variables.

Finally, by these extended lemmas, we can prove the SLT existence within the frozen network with a sufficiently large width.

**Theorem A.1** (SLT Existence in Frozen Networks). *Let $\mathcal{D}$ be input data, $f_T$ be a target network with depth $L$, and $f_S$ be a source network with depth $L + 1$, $\boldsymbol{\theta}_S$ be a parameter of $f_S$, and $p_l, q_l \in (0, 1)$. Assume that these networks have ReLU activation functions, and each element of $\boldsymbol{\theta}_S$ is uniformly distributed over $[-1, 1]$. Also assume that $\boldsymbol{\theta}_S$ is randomly pruned and locked with pruning ratio $p_l$ and locking ratio $q_l$ for each $l$-th layer, except for $0$-th and $1$-st layers. Then, with probability at least $1 - \delta$, there exists a binary mask $\boldsymbol{m}_S$ so that each output component $i$ is approximated as $\max_{\boldsymbol{x} \in \mathcal{D}} \|f_{T,i}(\boldsymbol{x}) - f_{S,i}(\boldsymbol{x}; \boldsymbol{\theta}_S \odot \boldsymbol{m}_S)\| < \varepsilon$ if*

$$n_{S,l} \geq C_l \log\left(\frac{C_l'}{\min\{\varepsilon_l, \delta/\rho\}}\right) n_{T,l}, \qquad (l \geq 1) \tag{11}$$

$$n_{S,0} \geq C_0 \log\left(\frac{C_0'}{\min\{\varepsilon_1, \delta/\rho\}}\right) d \qquad (l = 0) \tag{12}$$

*where $C_l, C_l' > 0$ are constants, and each $C_l$ includes $p_l$ and $q_l$, except for $l = 0$. Also, $\varepsilon_l$ and $\rho$ are as defined in Burkholz (2022a); Gadhikar et al. (2023).*

*Proof.* By following the proof procedure of Gadhikar et al. (2023) using our Lemma A.2 instead of Lemma 2.4 in Gadhikar et al. (2023), we can obtain the desired result as an extension of Theorem 2.5 in Gadhikar et al. (2023). $\qquad\square$

# B    Additional Experimental Results

## B.1    Accuracy of SLT within Dense ResNet-50

This section introduces the preliminary experiment used for determining the main experimental setup. Figure 7 compares the accuracy of SLTs within a dense ResNet-50 source at different SLT sparsity using ImageNet. The SLT with 80% sparsity is the most accurate, reaching 66.8% accuracy.

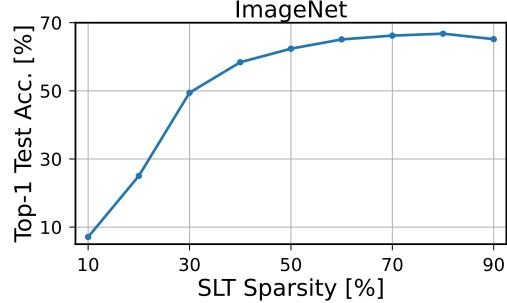

Figure 7: Accuracy comparison of SLTs of different sparsity within a dense ResNet-50 on ImageNet.

## B.2    Accuracy of SLT within Transformer Architectures

In this section, we search for SLTs within DeiT-S (Touvron et al., 2021), the Transformer architecture for vision tasks, using ImageNet. The experimental setup is same as in the DeiT paper, and we use `Edge-Popup` for finding SLTs.

As shown in Table 2, compared to the existing SLT search methods, we can find more accurate SLTs from the frozen network, even though the model size is the same or smaller. However, the SLT accuracies are significantly lower than that of the weight-trained network. While CNN and GNN maintain some SLT accuracy regardless of network pre-processing (i.e., non-frozen, pruning only, or frozen), as shown in Table 1, DeiT shows a significant decrease in SLT accuracy even without freezing. It suggests there are some problems specific to the Transformer architecture with respect to SLT. As mentioned in Section 5, SLT within Transformer architectures has not yet been established experimentally and theoretically, and investigating this degradation is left for future work.

Table 2: SLT accuracy comparison in the DeiT-S architecture.

| | | | DEIT-S & IMAGENET | | | |
|---|---|---|---|---|---|---|
| Method (Source Net.) | Weight Init. | Sparsity [%] | Pruning Ratio [%] | Locking Ratio [%] | Top-1 Test Acc. [%] | Model Size [MB] |
| Weight Training | KU | - | - | - | 78.4 | 83.59 |
| SLT (Dense) | SKC | 60 | 0 | 0 | 51.8 | 2.61 |
| SLT (Sparse) | SKC | 60 | 50 | 0 | 10.3 | 1.31 |
| **SLT (Frozen)** | SKC | 60 | 35 | 15 | **40.2** | **1.31** |
| **SLT (Frozen)** | SKC | 60 | 40 | 20 | **36.9** | **1.04** |

