# OpenReview forum: "Partially Frozen Random Networks Contain Compact Strong Lottery Tickets"
_TMLR — Accepted by TMLR_

### Review · Reviewer_JSuv · 2024-12-14

**Summary Of Contributions:**

This paper proposes a method to find strong lottery tickets (SLTs), which aims to increase the performance of SLTs at the desired sparsity level while compressing memory consumption used for parameter masks. The existence of SLTs is theoretically provided in the paper's setting. Experimental results for CNN and GNN are provided to support the strength of the proposed method.

**Audience:**

Yes

**Claims And Evidence:**

Yes

**Requested Changes:**

Please see the weakness parts.

**Strengths And Weaknesses:**

### Strengths
- Background and motivations are explained very well.
- The theoretical result for SLT existence is provided in the setting with locked parameters.
- Figures clearly represent the key points in the proposed idea.
- The experimental results are provided not only in simple toy models but also in various model and dataset settings, such as GNN and ImageNet.

### Weaknesses
My major concern is that the novelty compared to [Gadhikar+,2023] may be limited and whether the extension of the paper is sufficiently meaningful.

- The practical advantages from [Gadhikar+,2023] are unclear. Is it aimed at accelerating the finding of SLTs or reducing the memory consumption required for obtained SLTs? It is mentioned that the supermask can be efficiently compressed through entropy coding, but how significant are the practical benefits of making the supermask smaller?
- The specific calculation for the model sizes in the figures and table is unclear.
- The selection of the target sparsity and pruning ratio in Table 1 seems arbitrary. If the target SLT sparsity is very close to the pruning ratio, test accuracy drops significantly, as shown in Figure 4. This might allow the pruning-only baseline in [Gadhikar+,2023] to be artificially lowered. Figures 2 and 6 use model size as the horizontal axis, making the actual SLT sparsity level unclear and giving the impression that this aspect is hidden.
-  Regarding the theoretical analysis, by defining new target $z^\prime = z - \sum M^\prime_i X_i$, where the subtraction comes from the contribution of the locked parameters, wouldn't the results of [Gadhikar+, 2023] immediately lead to the conclusions? If not, it would be better to write about the difficulty of the analysis.

---

> ### Author Response · Authors · 2025-01-09
>
> Thank you for your valuable feedback and constructive comments. We would like to answer to your comments as follows:
>
> ---
> >The practical advantages from [Gadhikar+,2023] are unclear. Is it aimed at accelerating the finding of SLTs or reducing the memory consumption required for obtained SLTs?
>
> The practical advantage we focused on is not accelerating the finding of SLTs but the memory reduction required to use the obtained SLTs, as shown in the following sentences in the introduction and conclusion sections. Accelerating the finding of SLTs remains a future work:
>
> - This paper introduces such a novel method to reduce the memory cost of the optimized supermask without restricting the desired sparsity of SLTs to be searched for:
>
> - (Before revision) Additionally, the reduced number of parameters to be optimized may be exploited for training cost reduction, which remains for future work.
>
> - (After revision) Additionally, the reduced number of parameters to be optimized may be exploited for training cost reduction.
> Appropriate freezing may improve the SLT search efficiency, including preventing convergence to local low-precision solutions as experimentally confirmed in Section 4.2, and improving convergence speed through reduced parameters to be optimized.
> Investigations about the SLT search efficiency remain as future work.
>
> ---
>
> >It is mentioned that the supermask can be efficiently compressed through entropy coding, but how significant are the practical benefits of making the supermask smaller?
>
> In SLT-specialized HW, whether the supermask is encoded or not, inference speed and power consumption are affected by the number of off-chip memory accesses to obtain supermask information. Therefore, the smaller the supermask, the higher the speed and the lower the power consumption. (Note that if we have a very small supermask in advance, the encoded mask can also become very small in general.)
> Furthermore, the small supermask may allow us to avoid the entropy encoding/decoding process, which requires additional computation overhead, specifically on resource-constrained HWs.
> These are our motivations for making a supermask smaller, and we believe these advantages on the SLT-specialized HW side due to making the supermask smaller are practical benefits.
>
> ---
>
> >The specific calculation for the model sizes in the figures and table is unclear.
>
> As described in Section 4.3, the model size is calculated as the sum of the unfrozen partial supermask (1 bit/parameter) and the batch normalization parameters (32 bits/parameter).
> We added the model size description in section 4.5 as follows:
>
> - As described in Section 4.3, the model size is calculated as the sum of the unfrozen partial supermask (1 bit/parameter) and the batch normalization parameters (32 bits/parameter).
>
> ---
>
> >The selection of the target sparsity and pruning ratio in Table 1 seems arbitrary.
>
> Table 1 summarizes specific numerical results from the experiment section, so the choice of target SLT sparsity and pre-pruning ratio is based on the experiment section.
> In the experiment section, the SLT sparsity is set as the most accurate SLT sparsity in a dense network, as shown in the following sentence.
>
> - SLT sparsity is fixed to that of the best performing SLT found in a dense source network in Figure 5:
>
> Also, as shown in the following sentence in section 3.2, the pre-pruning ratio in the experiment section is determined based on the freezing ratio and SLT sparsity.
>
> - Consequently, hereafter, the proposed method sets the global ratios in order to position the frozen region center as close as possible to the desired SLT sparsity, and then sets the layer-wise ratios using EPL.
>
> We added the description of target SLT sparsity and pre-pruning ratio in section 4.5 as follows:
>
> - Also, as described in Section 4.2, the target SLT sparsity is fixed to that of the best-performing SLT found in a dense source network, and we determine the pre-pruning ratio in the sparse network as (SLT sparsity − 5)% for high compression.
>
> ---

---

> ### Author Response · Authors · 2025-01-09
>
> >If the target SLT sparsity is very close to the pruning ratio, test accuracy drops significantly, as shown in Figure 4. This might allow the pruning-only baseline in [Gadhikar+,2023] to be artificially lowered.
>
> Since we are interested in the accuracy of the SLTs compressed as much as possible obtained by each method, the pre-pruning ratio for the pruning-only case is determined as target sparsity - 5 % to compress the SLT as much as possible while allowing to search for SLTs.
> On the other hand, we evaluate freezing with a higher compression ratio than pruning only. This setting means we evaluate our method under more strict conditions than pruning-only from the compression ratio perspective.
> Note that if the pre-pruning ratio were set to a smaller value, SLT would not be compressed sufficiently. So, making such an evaluation setup inconsistent with the focus of our work.
> We added the following sentences about the pre-pruning ratio for the pruning-only case to section 4.3:
>
> - Note that since we are interested in the accuracy of the SLTs compressed as much as possible obtained by each method, the pre-pruning ratio for the pruning-only case is determined as target sparsity - 5 % to compress the SLT as much as possible while allowing to search for SLTs. On the other hand, we evaluate freezing with a higher compression ratio than pruning only. This setting means we evaluate our method under more strict conditions than pruning-only from the compression ratio perspective.
>
> ---
>
> >Figures 2 and 6 use model size as the horizontal axis, making the actual SLT sparsity level unclear and giving the impression that this aspect is hidden.
>
> The SLT sparsity is set as the most accurate SLT sparsity in a dense network, as shown in the following sentence in section 4.3 and section 4.4.
>
> - SLT sparsity is fixed to that of the best performing SLT found in a dense source network in Figure 5:
> - Since SLTs with 80% sparsity achieve the highest accuracy in a dense source ResNet-50 (for details, see Appendix B.1), we compare the three methods using 80% SLT sparsity.
>
> The SLT sparsity with each figure is also stated in the legend and captions of Figure 2 and 6.
>
> ---
>
> >Regarding the theoretical analysis, by defining new target $z'=z−\sum M_i'X_i$, where the subtraction comes from the contribution of the locked parameters, wouldn't the results of [Gadhikar+, 2023] immediately lead to the conclusions? If not, it would be better to write about the difficulty of the analysis.
>
> The results of Gadhikar et al., 2023 can be applied only when the target $z$ is deterministic and bounded by a constant.
> On the other hand, defining z′ in your way introduces a stochastic term $\sum M_i'X_i$, which makes $z'$ stochastic, and the bounds of $z'$ are not fixed at a constant.
> Consequently, directly substituting $z'$ into their theorems does not establish our lemmas and theorems.
> We added this description in the following sentences in the Appendix, emphasizing the analytical challenges addressed in our work.
>
> - These corollaries seem to be derived by taking $z' := z - \sum_{i=1}^{n}M_iX_i$ as the target of the Lueker (1998) and Gadhikar et al. (2023) corollaries. However, it cannot. Their corollaries need the target to be deterministic and have constant bounds. On the other hand, since $\sum_{i=1}^{n}M_iX_i$ is stochastic, $z'$ is a stochastic value, and the bounds of $z'$ are not constant. Therefore, $z'$ cannot be directly assigned to the conventional corollaries, and our approach is necessary for proofing the subset-sum approximation with frozen variables.
>
> ---
>
> Thank you for taking the time to read our paper. We hope our answers have addressed your concerns.

---

> > ### Comment · Reviewer_JSuv · 2025-01-19
> >
> > Thank you for your reply. My concerns have been resolved.
> > In particular, I am convinced by the discussion regarding off-chip memory access and that the (target sparsity - 5)% setting is determined as the highest compression while still enabling the identification of SLT.

---

### Review · Reviewer_1gAq · 2024-12-19

**Summary Of Contributions:**

The paper proposes a method for discovering Strong Lottery Tickets (SLTs) in neural networks by freezing parameters through random pruning and locking at initialization. It also provides preliminary theoretical evidence suggesting that SLTs can still be found within these frozen networks, given that the networks have sufficient width.

**Audience:**

Yes

**Broader Impact Concerns:**

No concerns.

**Claims And Evidence:**

Yes

**Requested Changes:**

Please see the weakness section.

**Some minor issues**:

Some sections feel redundant, particularly the introduction and the beginning of Section 3. I think it would be helpful streamline these parts to avoid repetition.

Certain sentences are unclear. For example, in Section 2.1, the sentence:

     "Such as this top-k, these methods determine the SLT sparsity to be explored as a hyperparameter explicitly or implicitly."

**Strengths And Weaknesses:**

**Strength:**
   * The paper addresses the issue that previous methods restricted search to suboptimal sparsity regions.
   * The proposed method demonstrates improved accuracy-to-model size trade-offs through empirical evaluation.
   * The evaluation is comprehensive, including tests across multiple architectures, various datasets, and three sparsity scenarios.

**Weakness**
   * Although the proposed method for finding SLTs in a frozen network should work with different search methods, the paper primarily experiments with Edge-Popup. Is there a specific reason for this limitation?
   * (Minor) While the authors acknowledge the lack of hardware implementation as a limitation, the paper could still be strengthened by providing quantitative results on storage reduction and inference efficiency through simulated experiments or analytical comparisons.
   * (Minor) It might be beneficial to include more discussion on how different locking ratios affect convergence speed and computational cost during the search process.

---

> ### Author Response · Authors · 2025-01-09
>
> Thank you for your insightful feedback and constructive suggestions.
> Below are our responses to each comment:
>
> ---
>
> >Although the proposed method for finding SLTs in a frozen network should work with different search methods, the paper primarily experiments with Edge-Popup. Is there a specific reason for this limitation?
>
> As you say, our method is not limited to Edge-Popup but can be applied to other search methods.
> However, other search methods, such as Gem-Miner [1] and ProbMask [2] is only experimentally clear that these methods mainly mitigate accuracy degradation in extreme sparsity settings (e.g., SLT sparsity > 95%), and they do not aim to find the global optimum in the network.
> In contrast, Edge-Popup has been empirically shown to search for various sparsities and can obtain accurate SLTs in the network, making it suitable for our goal of finding the most accurate SLT within some pre-processed networks. For this reason, this paper used Edge-Popup.
>
> [1] Sreenivasan et al., "Rare gems: Finding lottery tickets at initialization," NeurIPS'22
>
> [2] Zhou et al., "Effective sparsification of neural networks with global sparsity constraint," CVPR'21
>
> ---
>
> >(Minor) While the authors acknowledge the lack of hardware implementation as a limitation, the paper could still be strengthened by providing quantitative results on storage reduction and inference efficiency through simulated experiments or analytical comparisons.
>
> We agree that including such results would strengthen the paper. However, conducting simulations or detailed analytical comparisons requires additional resources beyond the current scope of this work.
>
> ---
>
> >(Minor) It might be beneficial to include more discussion on how different locking ratios affect convergence speed and computational cost during the search process.
>
> We agree that the effect of locking in searching for SLTs is an intriguing research direction.
> We added the following discussion about searching for SLTs and network freezing in the conclusion section.
>
> - Appropriate freezing may improve the SLT search efficiency, including preventing convergence to local low-precision solutions as experimentally confirmed in Section 4.2, and improving convergence speed through reduced parameters to be optimized. Investigations about the SLT search efficiency remain as future work.
>
> ---
>
> >Some sections feel redundant, particularly the introduction and the beginning of Section 3. I think it would be helpful streamline these parts to avoid repetition.
>
> We modified the beginning of section 3 to ensure it is concise and avoids unnecessary repetition.
>
> ---
>
> >Certain sentences are unclear. For example, in Section 2.1, the sentence: "Such as this top-k, these methods determine the SLT sparsity to be explored as a hyperparameter explicitly or implicitly."
>
> We deleted the relevant sentence. In addition, we checked for grammatical ambiguities.
>
> ---
>
> Thank you for taking the time to read our paper. We hope our answers have addressed your concerns.

---

### Review · Reviewer_TKUm · 2024-12-27

**Summary Of Contributions:**

- Compared to previous works, this paper proposes a method that extracts strong lottery tickets (STLs) from random networks whose parameters are not only pruned but also locked (namely, random parameters that are for sure part of the STL).
- They show that when applying the Edge-Popup method to these partially frozen random networks, the found STLs achieve better accuracy than on within dense or pruned-only random networks.
- Their method enables the memorization of fewer parameters compared to Sota methods.

**Audience:**

Yes

**Broader Impact Concerns:**

I have no concerns.

**Claims And Evidence:**

Yes

**Requested Changes:**

Minor comment: I think the color palette in the captions of Figures 2 and 6 differs from those in the legends.

**Strengths And Weaknesses:**

### Strengths:
- The paper is well-written and organized.
- The rationale behind the work is solid.
- The addressed topic is important for the model compression community.

### Weaknesses:
- The novelty is limited (compared to previous work, it seems the only improvement is the locking part of the source network), however the results are promising.
- The proposed method is tested on a limited number of architectures and datasets. I know the authors pointed it out as a limitation in the main paper, but it would be interesting to show some preliminary experiments on transformers-based architectures ().

---

> ### Author Response · Authors · 2025-01-09
>
> Thank you for your detailed feedback and helpful comments. We would like to answer to your comments as follows:
>
> ---
>
> >The novelty is limited (compared to previous work, it seems the only improvement is the locking part of the source network), however the results are promising.
>
> We appreciate your acknowledgment of our results.
> While we agree that the primary algorithmic distinction lies in introducing network locking, our work contains other contributions as follows: We show how pre-pruning and locking the network can significantly contribute to model compression in the SLT context. Furthermore, we provide theoretical proof related to this freezing approach. We believe these contributions show the novelty of our paper beyond the locking mechanism.
>
> ---
>
> >The proposed method is tested on a limited number of architectures and datasets. I know the authors pointed it out as a limitation in the main paper, but it would be interesting to show some preliminary experiments on transformers-based architectures ().
>
> We conducted preliminary experiments using the DeiT (Vision Transformer) architecture. The results and discussion are following. We included these contents in the Appendix.
>
> | Method (Source Net.) | Weight Init. | Sparsity [%] | Pruning Ratio [%] | Locking Ratio [%] | Top-1 Test Acc. [%] | Model Size [MB] |
> | :------------------- | :----------: | -----------: | ----------------: | ----------------: | ------------------: | --------------: |
> | Weight Training      | KU           | -            | -                 | -                 | 78.4                | 83.59           |
> | SLT (Dense)          | SKC          | 60           | 0                 | 0                 | 51.8                | 2.61            |
> | SLT (Sparse)         | SKC          | 60           | 50                | 0                 | 10.3                | 1.31            |
> | **SLT (Frozen)**     | SKC          | 60           | 35                | 15                | **40.2**            | **1.31**        |
> | **SLT (Frozen)**     | SKC          | 60           | 40                | 20                | **36.9**            | **1.04**        |
>
> - compared to the existing SLT search methods, we can find more accurate SLTs by searching from the frozen network, even though the model size is the same or smaller. However, the SLT accuracy is significantly lower than that of the weight-trained network. Since the accuracy of SLTs obtained from the dense network is also extremely low, we consider that this accuracy degradation is a specific problem of SLTs within the DeiT. As mentioned in Section 5, SLT within Transformer architectures has not yet been established experimentally and theoretically, and investigating this degradation is left for future work.
>
> ---
>
> >Minor comment: I think the color palette in the captions of Figures 2 and 6 differs from those in the legends.
>
> Thank you for pointing this out. We corrected the color.
>
> ---
>
> Thank you for taking the time to read our paper. We hope our answers have addressed your concerns.

---

> > ### Comment · Reviewer_TKUm · 2025-01-16
> >
> > Thank you for your answer and the preliminary results on a ViT. You mention that "we consider that this accuracy degradation is a specific problem of SLTs within the DeiT". What are these specific problems? I'm also wondering why the top-1 test accuracy for STL (Sparse) is so low, could it be that there is just a hyperparameter search problem?

---

> > > ### Author Response · Authors · 2025-01-22
> > >
> > > Thank you for your interest in our result.
> > > Below are our answers to your comment:
> > >
> > > ---
> > > > You mention that "we consider that this accuracy degradation is a specific problem of SLTs within the DeiT". What are these specific problems?
> > >
> > > First of all, note that we wrote “specific problem of SLTs within the DeiT” just because we performed experiments only with the DeiT model, but actually we intended more general problem specific to the Transformer architecture.
> > >
> > > Our experiments show that SLTs obtained from CNNs and GNNs are not significantly less accurate than trained-weight networks regardless of the network pre-processing (unfrozen, pruned, or frozen). In contrast, the Transformer architecture, including DeiT, has a significant overall drop in accuracy.
> > > Therefore, we consider there are some problems specific to the Transformer architecture, particularly attention mechanisms.
> > > Since the theoretical existence of SLTs in the attention mechanism is still an open question, we expect that proving their existence will lead to the finding of highly accurate SLTs within Transformers.
> > >
> > > For the sake of improved clarity, the sentence you commented has been modified as follows:
> > >
> > > - As shown in Table 2, compared to the existing SLT search methods, we can find more accurate SLTs from the frozen network, even though the model size is the same or smaller. However, the SLT accuracies are significantly lower than that of the weight-trained network. While CNN and GNN maintain some SLT accuracy regardless of network preprocessing (i.e., non-frozen, pruning only, or frozen), as shown in Table 1, DeiT shows a significant decrease in SLT accuracy even without freezing. It suggests there are some problems specific to the Transformer architecture with respect to SLT. As mentioned in Section 5, SLT within Transformer architectures has not yet been established experimentally and theoretically, and investigating this degradation is left for future work.
> > >
> > > ---
> > >
> > > > I'm also wondering why the top-1 test accuracy for STL (Sparse) is so low, could it be that there is just a hyperparameter search problem?
> > >
> > > It is actually unclear why the significant performance drop in SLT (sparse) occurs for Transformers.
> > > We suspect that the Transformer is incompatible with SLT, especially with SLT (sparse), based on following facts: 1) the performance of SLT within DeiT is decreased regardless of the pre-processing as shown in our experiments; 2) the robustness of the Transformer to pruning with high sparsity, has not been confirmed compared to CNNs (e.g., [1]).
> > > Also, as described in Section B.2, the hyperparameters used in DeiT experiments are essentially based on the paper of DeiT, while the SLT-specific hyperparameters (SLT sparsity, pre-pruning ratio, and locking ratio) are based on the settings of the experiment section.
> > > Appropriate adjustments to these hyperparameters may improve accuracy.
> > > However, as described in the above comment and limitation section, SLT has not yet been established for general Transformers experimentally and theoretically.
> > > So, the detailed investigation is beyond the scope of this paper.
> > > As noted in the limitation section, we leave it for future work to investigate SLTs within the Transformer architecture.
> > >
> > > [1] Kuznedelev et al., "CAP: Correlation-Aware Pruning for Highly-Accurate Sparse Vision Models," NeurIPS'23

---

### Decision · Action_Editor_r7Uz · 2025-01-25

**Recommendation:** Accept as is

**Comment:**

The reviewers appreciated the clarity of exposition. Although the incremental novelty relative to [Gadhikar et al., 2023] was raised as a concern, TMLR does not require a high degree of “novelty” if the submission provides new insights, solid evidence, and potential utility for readers. The partial-freezing strategy and accompanying experiments address a worthwhile niche in memory-constrained SLT discovery.

The authors’ rebuttals convincingly clarified their contributions, made improvements to the draft, and demonstrated that the claims are properly supported. The audience would likely learn from these findings and appreciate the new angle on “frozen” SLTs and their practical trade-offs.

**Audience:**

Shall be of broad interest

**Claims And Evidence:**

Overall, the submission’s key claim is that partial freezing of randomly initialized networks can reduce the memory size of supermasks used to obtain strong lottery tickets (SLTs), while still preserving (and sometimes improving) accuracy compared to SLTs found in dense or randomly pruned source networks. This claim is backed well by sound methodological development, comprehensive empirical results, and theoretical argument.